# In Site Soil Seed-Banks: Size, Composition and Persistence across Tropical Successional Stages

**DOI:** 10.3390/plants12152760

**Published:** 2023-07-25

**Authors:** Julieta Benitez-Malvido

**Affiliations:** Instituto de Investigaciones en Ecosistemas y Sustentabilidad, Universidad Nacional Autónoma de México (UNAM), Antigua Carretera a Pátzcuaro No 8701, Col. Ex-Hacienda de San José de la Huerta, Morelia 58090, Mexico; jbenitez@cieco.unam.mx; Tel.: +52-4434100951

**Keywords:** old-fields, persistence, plant growth-forms, primary forests, regeneration, secondary forests, Selva Lacandona

## Abstract

I investigated the size, composition and persistence of the seed-bank in primary forests, secondary forests and old-fields in southern Mexico. I also assessed the contribution of the seed-bank to regeneration relative to other propagule sources. In all habitats, I removed by hand all plants and litter and excluded the seed-rain. For one year, I counted the number of plant species (5–50 cm tall) emerged and grouped them into different growth-forms: trees, shrubs, palms, herbs, woody lianas, epiphytes and hemi-epiphytes. A total of 95 species emerged. The seed-bank size, composition and persistence showed strong variation among successional stages. Emergence was low for primary and secondary forests, but high for old-fields (19, 26, and 68 plants per m^−2^, respectively). Herbs were the most abundant in the seed-bank and palms the less. Time had a negative effect on seed-bank size in primary forests and old-fields; whereas for secondary forests size remained constant throughout the year. The number of emerged plants in different growth-forms changed significantly across time for all successional stages. Overall, the seed-bank provided a greater number of plants in old-fields relative to other propagule sources combined. The results showed that forest modification alters the input of propagules throughout the seed-bank for different plant growth-forms.

## 1. Introduction

The regeneration of tropical rain forest plants depends on different propagule sources [1,2,3]. Species contributing to forest regeneration can either be present in the aboveground vegetation (e.g., seedlings, saplings, resprouting of broken plants, and lateral growth of branches), in the soil seed-bank or in the seed-rain. The relative importance of these regeneration sources varies spatially and temporally depending on the life history of the species and on the magnitude of forest disturbance and successional stage [1,2,3,4,5,6,7,8,9,10].

Many species of plants rely on regeneration from seed-banks (dormant seeds in the soil). Seed-banks are a major factor in the persistence of certain species [3,7,8,9,10,11] and are particularly important in plant communities that are subjected to extreme environmental events or frequent disturbances such as: fire, seasonal drought, periodically flooded areas or agricultural fields [5,6,10,11,12,13,14]. Furthermore, seed-banks play a vital role in the regeneration of tropical rainforests by maintaining the ecological and genetic diversity of populations and communities [14,15,16]. Seeds stored in the soil of tropical forests are an important source of new recruits and can therefore influence the course of forest regeneration and secondary succession [7,8,9,10,11]. In general, the seeds of pioneer tree species tend to dominate soil seed-banks [10,13] and seeds from these species are the first to germinate following disturbances [2,10,13]. Factors that influence the abundance and species composition of soil seed-banks in tropical rainforests include the following: land use history, landscape composition and configuration, floristic composition of current and past standing vegetation, phenology, seed physiology and dispersal mode [5,6,8,10,12,13,17,18].

Contemporary land use in tropical areas has strong and long-term lasting effects on the density of seeds stored in the soil, in their germination capacity and on their persistence [5,6,8,9,10,12,13,19]. Persistent seed-banks are particularly important for species with life spans shorter than the average perturbation cycle (e.g., recurrent fires) in order to counter local extinction in areas that have been subjected to slash and burn agriculture and/or severe management for the growth of cattle pastures [5,8,9,10,13]. Persistent seed-banks also have the potential for restoring ecosystems following disturbances such as fires, alien plant invasion and logging. In some places, seeds could survive for decades in the seed-bank [8,9]. 

Given the high diversity in plant species and growth-form classes of tropical rainforests [12,20,21,22,23,24,25], I explored the relative contribution of the soil seed-bank as a source of forest regeneration and recovery for species of trees (including pioneer and shade tolerant species), palms, shrubs, herbs, epiphytes, hemiepiphytes and woody lianas across different successional stages, encompassing recent old-fields, secondary forests and primary forests [12]. Most studies have measured the relative contribution and persistence of the seed-bank on forest regeneration by collecting soil cores, taking them to a green house or laboratory and counting the number of seeds or emerging plants or by using mesh bags to study seed-bank persistence [18,19]. In contrast, this study assesses the relative contribution and persistence of the soil seed-bank present in a given place and time, on the basis of recruited plants in situ [3,6,12,16,20,21,22]. In other words, the soil seed-bank was investigated using direct propagule emergence from the mineral soil, under field conditions [3,4]. 

To evaluate the spatiotemporal seed-bank differences in species abundance and composition at the different successional stages, I measured plant emergence from the seed-bank, inhibiting seed input from the seed-rain and the lateral growth of nearby plants. I also investigated the persistence of the soil seed-bank for 12 months considering several growth-forms and species and therefore, functional traits [12]. In particular, I addressed three questions: (i) Does the relative size of the seed-bank as a propagule source differ according to successional stage and growth-form? (ii) Does the persistence of the seed-bank differ across successional stages and growth-form? (iii) Do differences in the standing above-ground understory vegetation observed among successional stages could be explained by differences in the traits of the seed-bank?

## 2. Results

The number of emerged plants from the soil seed-bank differed significantly among successional stages (χ^2^ = 154.8, df = 2, *p* < 0.01), time (χ^2^ = 46.3, df = 2, *p* < 0.01), growth-form (χ^2^ = 304.3, df = 4, *p* < 0.01), and in the habitat x time (χ^2^ = 15.3, df = 4, *p* < 0.01), habitat x growth-form (χ^2^ = 201.6, df = 8, *p* < 0.01), and the habitat x time x growth-form (χ^2^ = 38.2, df = 24, *p* < 0.05), interaction terms. 

### 2.1. Patterns of Plant Emergence

The annual number of plants emerged from the soil seed-bank was greater in old-fields than in secondary and primary forests (68, 26, and 19 plants per m^−2^, respectively). Herbs dominated the seed-bank with nearly 54 percent of the total number of emerged plants, followed by epiphytes and hemiepiphytes (EH, 15%), shrubs (12%), trees (10%), lianas (9%) and palms (0.1%) (Appendix A). The seed-bank of primary forests was dominated by herbs (37%) and EH (32%) (Figure 1); that of secondary forests by trees and lianas (27% each) (Figure 2); and that of old-fields by herbs (73%) and shrubs (12%) (Figure 3). 

### 2.2. Species Composition of the Seed-Bank

In total, 95 species were recorded from the seed-bank, of which 27 were herbaceous species; 23 were trees; 19 were lianas; 14 were shrubs; 11 were EH; and there was one palm species (Figure 1, Figure 2 and Figure 3; Appendix A). The species richness of all growth-forms pooled was similar among habitats, with 35 species recorded in primary forests, 33 species in secondary forests and 37 species in old-fields. Thirty-three percent of the species emerged were exclusive to primary forests and old-fields; and 24% to secondary forests (Appendix A). Secondary forests shared 4.2% of the species with primary forests and 6.3 with old-fields; and no species was shared between primary forests and old-fields or among the three successional stages. 

The majority of the species that emerged from the seed-bank were rare, represented by one individual. The two dominant species emerged from the seed-bank per successional stage were the herb, *Calathea machroclamys* (18 plants/5 m^2^) and the EH, *Monstera* sp. 1 (15 plants/5 m^2^), for primary forests; the tree *Rinorea humilis* (19 plants/5 m^2^) and the shrub *Acalypha diversifolia* (15 plants/5 m^2^), for secondary forests; and two herbaceous species for the old-fields, Loganiaceae sp. 1 (77 plants/5 m^2^) and Asteraceae sp. 1 (51 plants/5 m^2^). Woody species, including tree seedlings and lianas were important in primary forests (Figure 1); whereas trees, lianas and shrubs were important in secondary forests (Figure 2). In contrast, for old-field woody plants were very scarce (Figure 3). 

### 2.3. Seed-Bank Persistence and Successional Stage 

Time had a strong negative effect on the persistence of the seed-bank in primary forests (Figure 1) and old-fields (Figure 3); whereas for secondary forests, soil seed reserves remained constant during one year (Figure 2). Plant density of emerged individuals was always greater in old-fields. The standing soil seed-bank in primary forests was depleted by 85% annually, that of old-fields by 50%; whereas that of secondary forests by 18%. The number of plants that emerged per growth-form changed significantly for all successional stages. In primary forests, there was a significant decline in plant emergence for all growth-forms along time except for lianas (Figure 1). In contrast, for secondary forests there was a decrease for herbs across time and an increase in the number of lianas for the second inventory (Figure 2). For old-fields there was a decline in the emergence of EH and herbs across time and a significant increase in the emergence of shrubs in the second inventory (Figure 3). 

### 2.4. The Relative Contribution of the Seed-Bank to Regeneration

The contribution of the seed-bank to forest regeneration compared to the intact plots used in a previous study as a reference differed according to successional stage and growth-form (Table 1). These control plots represent the natural density of recruited plants (<50 cm tall) across successional stages [3,4]. Pooling all plants, the seed-bank in old-fields provided a significantly greater number of individuals relative to other propagule sources combined (i.e., seed-rain and seed and seedling-banks, and lateral growth of branches); whereas no significant difference was observed for primary and secondary forests (Table 1). The relative importance of the seed-bank, considering each growth-form, was greater for lianas, in secondary forests; for herbs, in primary forests and old-fields; and for EH, in old-fields. In contrast, for trees and EH, the combination of propagule sources appeared to be more important than the seed-bank in primary and secondary forests, respectively.

## 3. Discussion

Overall, regarding the three questions raised in this study, I found that the relative size of the standing soil seed-bank varied according to successional stage, species growth-form and across time; whereas the standing above-ground low vegetation did not explain composition of the seedling species emerged from the soil seed-bank [3,4,8]. This study only considered recently emerged seedlings discarding adult reproductive individuals that might have higher species contribution to the seedling bank throughout the seed-rain than the soil seed-bank [3,4]. 

It is known that many processes involved in tropical rainforest regeneration and succession (e.g., seed dispersal, seed germination, and seedling establishment, survival and growth) are subjected to different types and regimes of disturbance (biotic and abiotic). In this study, the results showed that the deposition and storage of seeds in the soil is modified in secondary forests and old-fields as compared to primary forests [3,4,5,6,8,10]. The soil seed-bank is one of the drivers of succession, and differentiation among these seed-banks may contribute heavily to what kind of successional pathways are followed in particular areas [6,9,12]. In this study, all growth-forms but palms presented seed banks, but their size, species composition and persistence differed greatly among successional stages [6,9,12]. Overall, other propagules sources (i.e., seed-rain, sprouts, seedling-banks) may contribute to a greater extent to forest regeneration in primary and secondary forests; whereas in old-fields, the seed-bank appeared to be a major source of propagules [3,4,9,26,27,28,29,30,31] (Table 1).

### 3.1. Size of the Seed-Bank and Successional Stage

Local climate, species composition, fecundity of local plants, seed size, seed dormancy, seed dispersal and predation, season, and growth-form may explain the variation in seed emergence and soil seed-bank size and composition among successional stages [5,6,11,16,29,30,31,32,33]. The seed-bank in primary forests and old-fields was dominated by herbaceous and EH species, whereas for secondary forests by woody species (trees, shrubs and lianas). The low relative abundance of tree species and the high relative abundance of herbs have also been observed in other studies of soil seed-banks and forest succession [11,19]. Plots were located in alluvial terraces that were sporadically flooded, damaging the understory vegetation and creating tree fall gaps. Herbaceous and climbing plants may benefit from this kind of disturbance. Furthermore, late-successional tree species in primary forests often do not have stored seed-banks, as tree species produce seeds that rapidly germinate conforming the advanced regeneration community (i.e., seedlings and saplings in the shaded understory) [3,4,14], which may explain their low density in this study. Late-successional tree species usually have persistent seedling-banks and large seeds that cannot be dispersed easily over long distances, facts that may reduce forest regeneration from the seed-bank if the forest is destroyed. It is likely that the temporal patterns of abundance and composition in the seed-bank of secondary forests would have differed if I had used isolated sites and not those bordering primary forests, which might have increased seed dispersion and storage in the soil [5,34,35,36,37]. 

Contrary to other studies [19], tree species that were represented in the soil seed-bank were mostly late successional species [3]. This result could be a sampling artifact, however. The sampling for new emerged plants every four months could have underestimated the total emergence by a significant factor because early successional tree seedlings may emerge and die quickly [28], and those individuals that emerged and died between censuses were inevitably absent from the sample. Furthermore, the absence of early successional tree species may be simply the result of a mismatch between their fruiting phenology and the time when the standing seed-bank was sampled [19]. The time period over which data were collected may have affected the patterns of species abundance and distribution among successional stages [16,22].

The conditions in old-fields may impose limitations on colonization by woody species (e.g., reduce dispersal and physiological stress), whereas herbs and EH may have physiological adaptations to establish under the extreme conditions present in old-fields and may have persisted in the soil seed-bank after forest clearance [5]. Furthermore, in old-fields seed-bank size and composition, could be affected by heavy grazing, and by insect and fungal pathogens that may kill seeds [32]. In Manaus, Brazil, fungal attack to seeds was greater in old-fields dominated by *Vismia* sp. than in primary forests (G. Ganade, pers. inf.). Although most of the species that emerged from the seed-bank in old-fields were herbs and/or ruderals, they can develop a vegetative cover that may contribute to the reduction of soil erosion and once established may provide adequate microhabitats for the establishment of certain tree species facilitating forest succession [3,4]. 

### 3.2. Persistence of the Seed-Bank across Successional Stages

The existence of a seed-bank depends on the time allowed for seed accumulation; habitat accessibility to dispersing seeds, the type of disturbance, dispersal capability of colonizing species, the history of species distribution in the area and the life traits of the different growth-forms and species [5,12,16,27,28,33,34,35]. Seed-bank decline was faster for primary forests, followed by pastures and secondary forests. Apparently, primary forests hold species that lack seed-banks or species with short living seed-banks and/or fast germination; old-fields hold species with an intermediate life span seed-bank, whereas secondary forests species with long living seed-banks, indicating that the latter most probably hold many plant species that depend on the seed-bank for regeneration [3,16]. 

In primary forests, the rate of emergence from the standing seed-bank declined rapidly, probably because the most typical shade-tolerant forest plant species do not accumulate a persistent seed-bank. The rapid decline might be compensated by continuous seed input throughout the seed-rain and by the seedling bank [4,14,34,35], maintaining high understory plant diversity [34,35,36,37]. Secondary forests might as well have continuous local and external (surrounding primary forests) seed inputs (seed-rain) in addition of plant species with persistent seed-banks [3,5,34,35,36]. Finally, old-fields may hold long lived seed species in the seed-bank but a limited input from the seed-rain [3,4,5]. The existence of a persistent seed-bank may lead to a temporal rescue effect, where the extinction of a plant species is prevented through survival in the seed-bank [9,12,16,37]. Secondary forests showed long-term persistent seed banks and thus are buffered against years of poor seed production and/or seedling survival. However, seed-banks are major factors in the persistence of several woody weedy species as well [8,11,13]. 

### 3.3. Standing Vegetation and Soil Seed-Banks

Many studies had shown a poor correlation between species in soil seed-banks and those in the standing vegetation [15,16,19]. More than 80% of the species present in the standing vegetation were absent from the soil seed-bank in this study. It has been suggested that differences in species composition between the seed-bank and the standing vegetation could be the outcome of factors such as extreme events (fire, floods, drought, predation, etc.), which can destroy the soil seed bank and the standing vegetation; the season when the study was initiated [18], which may influence the incidence (presence or absence) of some species (e.g., ephemerals); on germination conditions; and on seed viability and dormancy. Furthermore, by studying the seed-bank present in a particular place and time using very small plants (≤50 cm tall), I assumed that most individuals (except for some herbs and EH) in the standing vegetation were not in their reproductive stage and hence did not contribute to the accumulation of seeds in the soil. The low similarity between the soil seed-bank and the standing vegetation may also imply that many species were lacking in the seed-bank because species turnover and accumulation over space and time was prevented with the experimental manipulation used in the study [16,28]. 

Overall, the study showed that, depending on the successional stage, the role of the seed-bank as a source of propagules changes rank in relation to plant growth-form and species. Recovery of degraded tropical areas could be more effective with a better understanding of regeneration processes and if information on soil seed-bank status of non-tree plant species is incorporated into tropical rain forest restoration programs [9,12,16]. 

## 4. Materials and Methods

### 4.1. Study Site 

The study was conducted at the Montes Azules Biosphere Reserve (MABR), Chiapas, southeastern Mexico (16°06′ N, 90°56′ W, 120 m elev.). The MABR is within the Selva Lacandona region that comprises part of Guatemala in Central America [23]. The original vegetation type is lowland tropical rainforest, reaching up to 40 m in height in alluvial terraces. There are around 4000 species of vascular plants [23,24]. Maximum and minimum annual temperatures are 31.8 °C (April–May), and 18 °C (January–February), respectively. The mean annual precipitation is 3000 mm, seasonally distributed along the year [25]. 

The landscape of the region includes forest patches of a variety of sizes, old-fields used for cattle and secondary forests of various ages. I searched for two areas (as experimental replicates) in alluvial terraces corresponding to each of three successional stages, including the following: (1) primary continuous old-growth forests, (2) secondary forests and (3) old-fields. I define secondary forests as those originating after complete forest clearance, as a consequence of human impact, and that regenerate after subsequent abandonment [38]. All study sites were at least 100 m apart. The secondary forests were bordering primary forests on one of their sides, were ca. 10 years old, about 8 ha in area, and with trees up to >15 m tall dominated by *Cecropia peltata*; while the old-fields were both used for cattle grazing, 10 ha in area, cleared 15 years before this study and have been abandoned for five years prior to this study. Old-fields were between 120–500 m from the nearest forest patch, and were wire fenced to avoid cattle trampling. The vegetation in the old-fields was dominated by sedges (*Scleria pterota*, *Cyperus* sp. and *Fymbristillys* sp.) and grasses (*Paspallum* sp., *Panicum* sp. and *Axonopus* sp.) presenting sparse woody species less than 3 m tall [3,4]. 

### 4.2. Sampling Design

Each successional stage has two replicates corresponding to different sites across the study landscape. To sample the seed-bank, five parallel, 100 m transects (20 m apart), were placed within each of the three successional stages [3,4]. For each site, 10, 0.50 m × 0.50 m permanent plots were randomly positioned along transects, giving a total of 20 plots per successional stage (10 plots × two replicates) [3,4]. The permanent plots were randomly distributed across transects and therefore, some transects had no plots along them. I decided to use small plots (0.25 m^2^) to facilitate experimental manipulation and because many small samples distributed over a large area provide greater accuracy and precision for estimating species richness of the soil seed-bank [16,19]. To evaluate the size, composition and persistence of the soil seed-bank, for each plot, the litter layer and vegetation (≤50 cm tall), roots, woody debris and stones were removed by hand and the surface mineral soil exposed. I decided to remove the litter layer because of the strong differences in litter quality and quantity among successional stages that may affect seed germination and plant emergence and to ensure that all plants emerged directly from the mineral soil. Furthermore, soil disturbance should activate small-sized dormant seeds and may be a factor in facilitating plant emergence [3,4]. Because all the upper organic layer was removed, I only considered the soil seed-bank that would correspond to seeds accumulated weeks or months previous to the start of the study.

To reduce vegetation interference and edge effects in the small plots, the vegetation 50 cm around all the experimental plots (the area where I recorded plants) was also removed [3,4]. To prevent seed input from above (seed-rain) and seed and plant predation, each plot was completely covered with a 0.5 mm transparent mesh mounted in a 40 cm tall wire frame fixed in the soil (Figure 4). The use of the mesh could have affected light income. Most seed-bank studies germinate seeds in brighter light conditions and it is known that many seeds will have no or reduce germination in low red/far red ratios. However, some light-demanding species with soil seed-banks also have advanced recruits under the closed canopy [3,26,32,33]. In the present study, more plants died within meshed than within control plots (un-meshed); however, I was considering plant emergence, not survival and growth [3,4].

Seed densities were estimated from the number of plants emerged within the 60 0.25 m^2^ plots. Emergence (in-growth of new plants) was recorded in all plots every four months over one-year (Figure 4). For each inventory, every newly emerged plant was labelled, identified to the lowest possible taxonomic group, and classified into different growth-forms: trees, shrubs, palms, herbs (terrestrial individuals including Pteridophyta), woody lianas and epiphytes and hemi-epiphytes, and identified them to the lowest possible taxonomic group. I pooled epiphytes and hemi-epiphytes (hereafter referred as EH) because some species (i.e., *Anthurium flexile*, *Philodendron inacuelaterum*, *Syngonium podophyllum*) have shown to change in growth habit depending on habitat and ontogeny. In the case of grasses and sedges, only when an entire tussock fell inside the plots, it was counted as an individual [4]. It is likely that this method underestimates true seed-bank size, especially for those species that require high light levels for germination and establishment and for those that die between inventories that were inevitably lost in the census. It is very difficult to estimate the magnitude of this bias because the lifespan of new recruits is poorly understood. 

To know the contribution of the seed-bank to forest regeneration relative to other propagule sources, I used data from a previous study in which the plants were recruited into intact control plots of the same size. These control plots were established at the same time and in the same study sites of the present study [3,4]. Due to the very low number of plants and species emerged from the seed-bank, I pooled the data of the two sites per successional stage for the analysis. Although the experiment was designed to avoid seed and plant predation, I observed small leaf damage on some recruits (<5% of leaf area). The experiment was set-up in September 1996.

### 4.3. Statistical Analysis 

Data were analyzed with Generalized Linear Models using the GLIM statistical package [39,40]. For all cases a Poisson error and a log-link function were used as indicated for cases of count dependent variables. In these models, the amount deviance explained by the independent variables approximates a χ^2^ distribution [39,40]. Deviance measures the discrepancy between the data and the fitted values. The change in deviance when a new term is fitted is a measure of the adequacy of that term. In the case in which a term was found to be significant the individual levels of the term (i.e., successional stage) were compared using t-tests [39]. The density and persistence of the seed-bank in different growth-forms across time was analyzed using the amount of deviance explained by three factors: successional stage, time, and growth-form, and the interactions among these factors. Finally, to test the significance of the difference in plant recruits at intact plots, relative to those recruited from the seed-bank, I used chi-square tests. Palms were not considered in the analyses as only a single individual emerged in old-growth forests during the course of the study.

## Figures and Tables

**Figure 1 plants-12-02760-f001:**
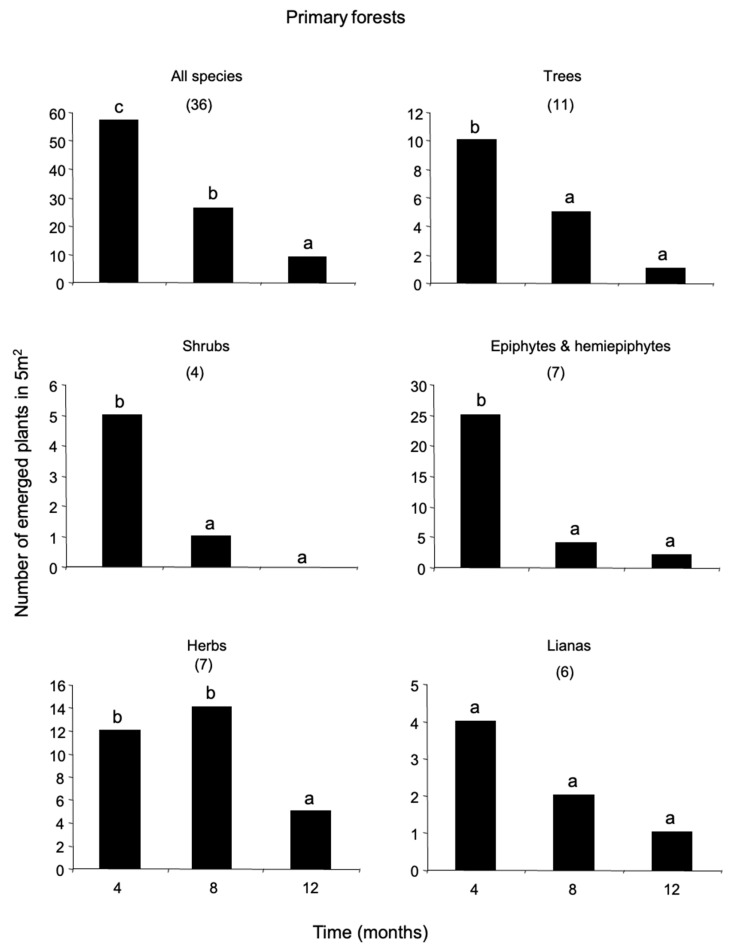
Plants emerged (≤50 cm tall) from the standing seed-bank in primary forests in southeast Mexico. Plant emergence was recorded every four months during one year. The figure shows the annual rates of seed-bank decline within plant growth-forms. Numbers in parenthesis indicate the total number of species recruited in different growth-forms. Bars sharing different letters are significantly different (*t* ≥ 2).

**Figure 2 plants-12-02760-f002:**
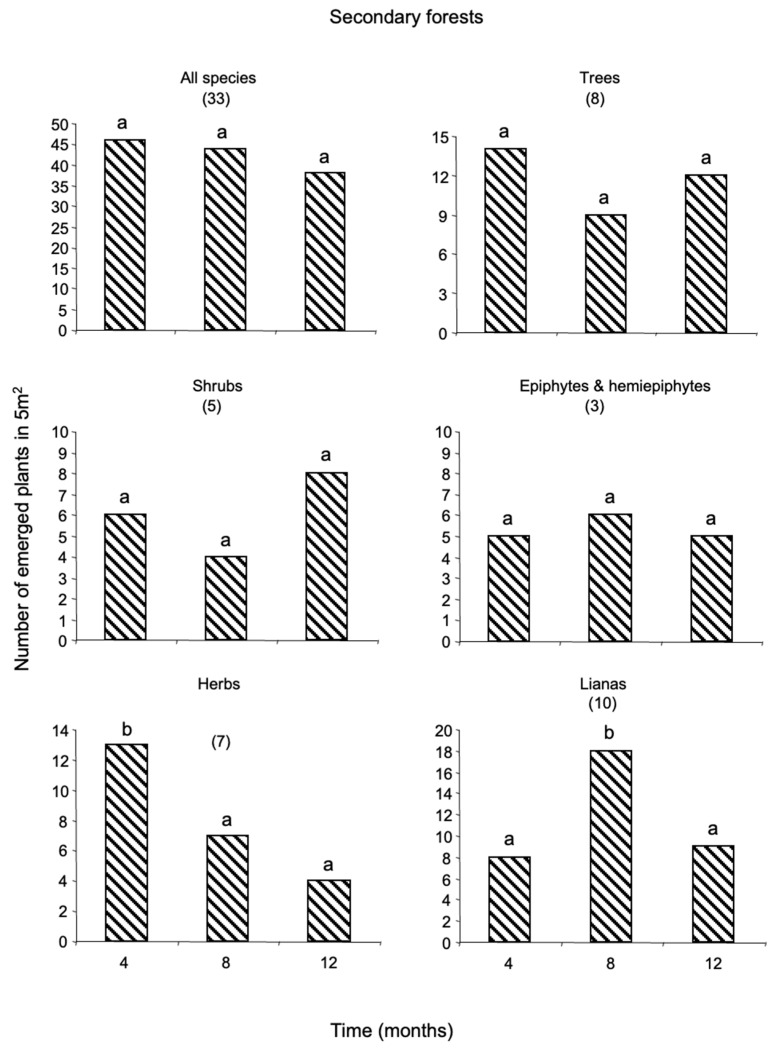
Plants emerged (≤50 cm tall) from the standing seed-bank in secondary forests in southeast Mexico. Plant emergence was recorded every four months during one year. The figure shows the annual rates of seed-bank decline within plant growth-forms. Numbers in parenthesis indicate the total number of species recruited in different growth-forms. Bars sharing different letters are significantly different (*t* ≥ 2).

**Figure 3 plants-12-02760-f003:**
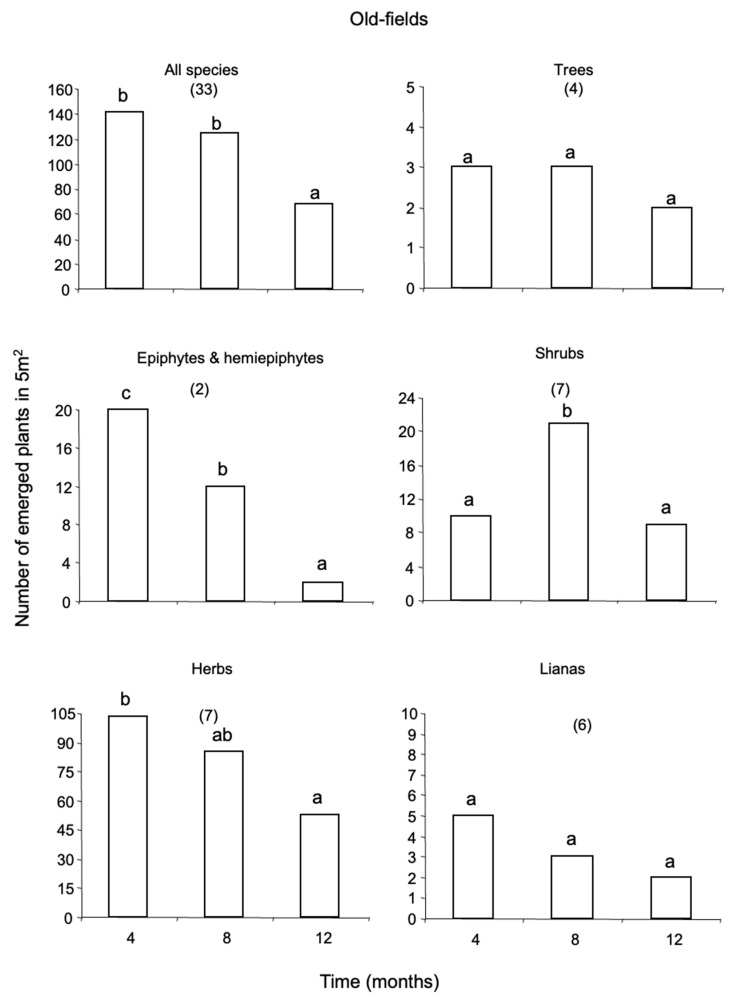
Plants emerged (≤50 cm tall) from the standing seed-bank in old-fields at Southeast Mexico. The figure shows the annual rates of seed-bank decline within plant growth-forms. Numbers in parenthesis indicate the total number of species recruited in different growth-forms. Bars sharing different letter are significantly different (*t* ≥ 2).

**Figure 4 plants-12-02760-f004:**
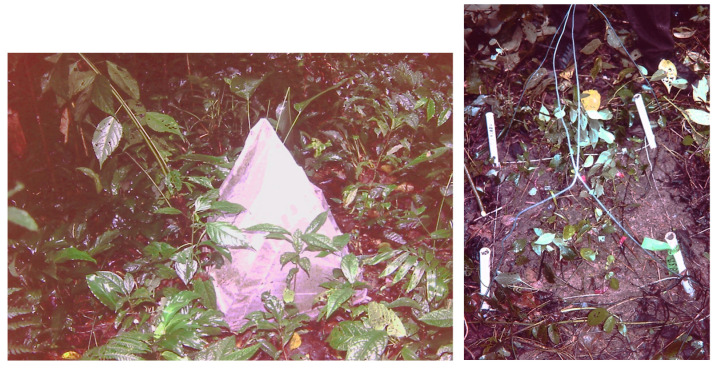
Experimental 25 cm^2^ plots, one covered with a transparent mesh to record plant recruitment from the soil seed-bank, exclusively. The other plot is uncovered to show the emerged plants. Plots were covered to record the recruited plants and classify them into different growth-forms and species. Meshed plots were used to exclude the seed-rain and to evaluate the contribution of the seed-bank to forest regeneration. Modified after [3].

**Table 1 plants-12-02760-t001:** Density of plants (≤50 cm tall) recruited in primary forests, secondary forests and old-fields in southeast Mexico through the seed-bank (SB) and those in control (C) plots with no experimental manipulation [3,4]. Numbers represent the plant density (no./5 m^2^) per successional stage in different growth-forms emerged and/or recruited during one-year.

Growth-Form and Successional Stage		Recruitment Mode		
		Control Plots ^1^	Seed-Bank	SB:C
All species				
	Primary forests	83	97	1.17
	Secondary forests	118	128	1.08
	Old-fields	162	340	2.10 **
Trees				
	Primary forests	33	16	0.48 *
	Secondary forests	31	30	0.97
	Old-fields	7	8	1.14
Shrubs				
	Primary forests	2	6	3.00
	Secondary forests	14	23	1.64
	Old-fields	32	40	1.25
Lianas				
	Primary forests	13	7	0.54
	Secondary forests	13	35	2.69 **
	Old-fields	5	10	2.00
Herbs				
	Primary forests	6	36	6.00 **
	Secondary forests	30	24	0.80
	Old-fields	104	248	2.38 *
Epiphytes and Hemi-epiphytes				
	Primary forests	20	31	1.55
	Secondary forests	29	16	0.55 **
	Old-fields	14	32	2.43 **

^1^ The SB:C ratio is the ratio of the number plants emerged from the seed-bank (SB) and those recruited in control plots (C) during one-year. These control plots represent the natural density of the understory vegetation (<50 cm tall) across successional stages. With chi-square analysis, SB:C values significantly different from 1.0 are * *p* < 0.05 and ** <0.01. Those not marked are not significant. Data of control plots are from [4].

## Data Availability

The data presented in this study are available in file: plants-12-02760-supplementary and contacting the author.

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
