# Peer review of "In Site Soil Seed-Banks: Size, Composition and Persistence across Tropical Successional Stages"

_plants, 2023, doi:10.3390/plants12152760_

Round 1

Reviewer 1 Report

This manuscript aims to find out the effects of soil seed banks on the regeneration of the forests. The design of this experiment is good, and the results can support the conclusion of this study. Generally, the writing of this paper is good.

Author Response

The reviewer suggests to improve the introduction and the explanation of the experimental methods. I have made the corrections

Reviewer 2 Report

Review on: In Site Soil Seed-Banks: Size, Composition and Persistence Across Tropical Successional Stages, by Julieta Benitez-Malvido.

The author examined the seed-bank of a tropical forest in three successional stages. Size, species composition, form of life trough the time were analyzed. All characteristics of seed-bank differed among successional stages, and seedling species composition varied across time. The author concludes disturbance affected the seed bank, but composition of standing vegetation did not explain composition of seedlings from seed-bank.

This paper is a valuable contribution to the journal. In general, it is well-written, and the experiments and analysis were well-performed. However, there are some issues that need to be addressed before the paper can be accepted for publication.

It’s not clear how many replicates there were, or how many sites per forest condition. Moreover, which was the control in table 1?

77-79: How the third question was assessed? Only, do you compare the number of different species between standing species and seedlings from seed-bank?

Why the “y” axis is in 5 m2/y?, I think it’s better to express this amount in 1 m2/y. In results, you use plants/m2. Data in figures and text should be aligned.

It’s a good idea to remember questions or hypothesis into discussion section. It helps to follow the reading and to have a better understanding of what the author is trying to tell us. Hypotheses were confirmed or rejected?

205-208: It should be very informative to know when the sampling was done, and when seed release occurs. however, if all the upper organic layer was removed, the seed-bank would correspond to previous years. Maybe, you could explore this result in terms of recalcitrance of seeds.

267-271: Instead a conclusion, this looks like a suggestion to take into account in this paper.

Discussion should improve with more recent literature and a better analysis/discussion of those similar/contrasting studies.

298-300: This is not clear. Why a total of 20 plots? There were five transects with four plots each one?

339-341: I don't understand where these data were used.

Minor suggestions

175-177: More clarity is needed in this sentence.

 Error bars in Figures (1-3)

Homogenize figures (1 – 3)

Title of Fig. 4 in editable “text”, currently text is part of the figure.

Author Response

Please find attached my responses (R:) in italics to the questions and suggestions raised by Reviewer 2. Ne adds and edits to the new version of the manuscript are marked in yellow.

Best regards

Julieta Benítez-Malvido

Review on: In Site Soil Seed-Banks: Size, Composition and Persistence Across Tropical Successional Stages, by Julieta Benitez-Malvido.

The author examined the seed-bank of a tropical forest in three successional stages. Size, species composition, form of life through the time were analyzed. All characteristics of seed-bank differed among successional stages, and seedling species composition varied across time. The author concludes disturbance affected the seed bank, but composition of standing vegetation did not explain composition of seedlings from seed-bank.

This paper is a valuable contribution to the journal. In general, it is well-written, and the experiments and analysis were well-performed. However, there are some issues that need to be addressed before the paper can be accepted for publication.

R: Thank you!

It’s not clear how many replicates there were, or how many sites per forest condition. Moreover, which was the control in table 1?

R: I rewrote this part to make it clearer. There were two sites per forest condition; the number of plots per forest condition were 20; in total I sampled seedling emergence from 60, 0.25 m2, plots.

R: The number of seedlings from control plots, come from another study which reference is: Benítez-Malvido, J. Effect of low-vegetation on the recruitment of plants in tropical successional successional stages. Biotropica 2006, 38:171-182. I rewrote this section to make it clearer.

This is indicated at the Table 1, footer: 1The SB:C ratio is the ratio of the number plants emerged from the seed-bank (SB) and those recruited in control plots (C) during one-year. With chi-square analysis, SB:C values significantly different from 1.0 are * P < 0.05 and ** <0.01. Those not marked are not significant. Data of control plots is after Benítez-Malvido (2006).

77-79: How the third question was assessed? Only, do you compare the number of different species between standing species and seedlings from seed-bank?

R: The third question raised was the following: “Do differences in the standing above-ground understory vegetation observed among successional stages could be explained by differences in the traits of the seed-bank?”

Yes, I addressed this question by comparing the density and species richness of seedlings between the standing vegetation using data from a previous study with those seedlings emerging from the seed-bank.

Why the “y” axis is in 5 m2/y?, I think it’s better to express this amount in 1 m2/y. In results, you use plants/m2. Data in figures and text should be aligned.

R: Done. I kept the density of emerged plants as the number of seedlings recruited in 5 m2/y in the figures and in the text. This is because I used the total number of plants emerged in one year within 20, 0.25 m2 plots, per forest condition.

It’s a good idea to remember questions or hypothesis into discussion section. It helps to follow the reading and to have a better understanding of what the author is trying to tell us. Hypotheses were confirmed or rejected?

R: I have followed this suggestion at the beginning of the discussion section.

205-208: It should be very informative to know when the sampling was done, and when seed release occurs. however, if all the upper organic layer was removed, the seed-bank would correspond to previous years. Maybe, you could explore this result in terms of recalcitrance of seeds.

R: Thank you for this observation. I have added this comment under the discussion section.

267-271: Instead a conclusion, this looks like a suggestion to take into account in this paper.

Discussion should improve with more recent literature and a better analysis/discussion of those similar/contrasting studies.

R: Done. I have added new relevant literature and made some changes in the discussion.

298-300: This is not clear. Why a total of 20 plots? There were five transects with four plots each one?

R: I have rewritten this section to make it clearer.  There were two sites per three forest conditions. In each site, five 100 m long parallel transects were positioned; transects were separated by 20 m. Ten plots for the seed-bank experiment, were positioned randomly across the five transects per forest condition in each site (60, 0.25 m2 plots, in total).

339-341: I don't understand where these data were used.

R: I have explained clearer. The data come from the meshed plots and from the control plots. Please see:

Benítez-Malvido, J.; Martínez-Ramos, M. ; Ceccon, E. Seed rain vs. seed bank, and the effect of vegetation cover on the re-cruitment of tree seedlings in tropical successional vegetation. Diss. Bot. 2001,346, 185-203.

Benítez-Malvido, J. Effect of low-vegetation on the recruitment of plants in tropical successional successional stages. Biotropica 2006, 38:171-182.

Minor suggestions

175-177: More clarity is needed in this sentence.

R: Done

Error bars in Figures (1-3):

R: Not needed, the figures represent the total number of emerged seedlings per forest condition

Homogenize figures (1 – 3)

R: Done

Title of Fig. 4 in editable “text”, currently text is part of the figure.

R: Done

Reviewer 3 Report

My comments are on the attached file

Author Response

I followed all the reviewer´s recommendation and added a Table indicating the species recorded and their growth form.
